# Partial Replacement of Fishmeal with Poultry By-Product Meal in Diets for Coho Salmon (*Oncorhynchus kisutch*) Post-Smolts

**DOI:** 10.3390/ani13172789

**Published:** 2023-09-01

**Authors:** Hairui Yu, Min Li, Leyong Yu, Xuejun Ma, Shuliang Wang, Ziyi Yuan, Lingyao Li

**Affiliations:** 1Key Laboratory of Biochemistry and Molecular Biology in Universities of Shandong (Weifang University), Weifang Key Laboratory of Coho Salmon Culturing Facility Engineering, Institute of Modern Facility Fisheries, College of Biology and Oceanography, Weifang University, Weifang 261061, China; liminyjsks@163.com (M.L.); leyong618@gmail.com (L.Y.); shuliang.wang@163.com (S.W.); moonn828@163.com (Z.Y.); 980714742@163.com (L.L.); 2Shandong Collaborative Innovation Center of Coho Salmon Health Culture Engineering Technology, Shandong Conqueren Marine Technology Co., Ltd., Weifang 261108, China; 3Weifang Marine Development and Fisheries Bureau, Weifang 261000, China

**Keywords:** biochemical tissue parameters, fish meal, growth performance, *Oncorhynchus kisutch*, poultry by-product meal, replacement

## Abstract

**Simple Summary:**

As an excellent but most expensive source of protein, fish meal (FM) is commonly used in 30–60% of the aquafeed for salmon fish species including coho salmon. However, the high cost and unstable supply of FM led it no longer to be considered as a sustainable protein source for aquafeeds. Accordingly, alternative protein sources with low cost and high availability have become a research hotspot in aquafeed for carnivorous species. Poultry by-product meal (PBPM), one of the rendered animal protein sources, can be used as protein substitute of FM in aquafeed due to its economical availability and high protein content and growth promotion effects in comparison to plant proteins. Until now, little information has been available on evaluating the effects of replacing FM by animal proteins including PBPM for coho salmon. In this study, the effects of PBPM instead of FM on growth, muscle composition, and tissue biochemical indexes of coho salmon were investigated. The results showed that the growth performance, feed utilization, muscle composition, serum biochemical indices, and liver antioxidant enzyme activities of coho salmon were negatively affected by high dietary inclusion level of PBPMs, and the optimum substitution level was evaluated based on the specific growth rate (SGR).

**Abstract:**

The present study evaluated the effects of partially substituting fish meal (FM) with poultry by-product meal (PBPM) on the growth, muscle composition, and tissue biochemical parameters of coho salmon (*Oncorhynchus kisutch*) post-smolts. Five isonitrogenous (7.45% nitrogen) and isoenergetic (18.61 MJ/kg gross energy) experimental diets were made by substituting 0%, 10%, 20%, 40%, and 60% FM protein with PBPM protein, which were designated accordingly as PBPM0 (the control), PBPM10, PBPM20, PBPM40, and PBPM60, respectively. Each diet was fed to triplicates of ten post-smolts (initial individual body weight, 180.13 ± 1.32 g) in three floating cages three times daily (6:50, 11:50, and 16:50) to apparent satiation for 84 days. Both specific growth rate (SGR) and protein efficiency ratio did not differ significantly (*p* > 0.05) among the control, PBPM10, and PBPM20 groups, which were remarkably (*p* < 0.05) higher than those of the PBPM40 and PBPM60 groups. Feed conversion ratio varied inversely with SGR. The PBPM replacement had no remarkable effects on the morphological indices and proximal muscle components. The control and PBPM10 groups led to significantly higher muscle contents of leucine, lysine, and methionine than groups of higher PBPM inclusion. The groups of PBPM40 and PBPM60 obtained significantly (*p* < 0.05) higher serum alanine aminotransferase and aspartate aminotransferase activities than the control and low PBPM inclusion groups. The control group had significantly higher albumin and total cholesterol contents than the groups with PBPM inclusion. The control group had significantly higher triglycerides content than the PBPM60 group. The PBPM60 group had significantly lower contents of high-density lipoprotein, low-density lipoprotein, and total protein than the control and PBPM10 groups. The high PBPM replacement level up to 40% and 60% had adverse effects on hepatic malondialdehyde levels. The catalase and superoxide dismutase activities were not affected by low PBPM inclusion, but significantly decreased in high-PBPM-inclusion groups. Based on broken-line regression analysis of SGR and PER, the optimum dietary PBPM replacing level was evaluated to be 16.63–17.50% of FM protein for coho salmon post-smolts.

## 1. Introduction

In the past several decades, the aquaculture of precious economic fish species has expanded rapidly and developed into a large-scale enterprise in China. Modern fish production in intensive aquaculture relies on formulated feed, which is the single largest cost of the production process and determines the productivity and profitability of aquaculture. Fish meal (FM), an excellent but most expensive source of protein, is commonly used in 30–60% of the aquafeed for predatory fish [1,2]. However, the high cost and unstable supply of FM led it no longer to be considered as a sustainable protein source for aquafeeds. Accordingly, alternative protein sources with low cost and high availability have become a research hotspot in aquafeed for carnivorous species [3,4,5,6,7,8,9].

Poultry by-product meal (PBPM), one of the rendered animal protein sources, can be used as a protein substitute of FM in aquafeed due to its economical availability and high protein content and growth promotion effects in comparison with plant proteins [10,11,12]. Many studies have investigated FM substitution with PBPM in diets of African catfish (*Clarias gariepinus*) [13], dourado (*Salminus brasiliensis*) [14], Florida pompano (*Trachinotus carolinus* L.) [15], golden pompano (*Trachinotus ovatus*) [16], humpback grouper (*Cromileptes altivelis*) [17], Pacific white shrimp (*Litopenaeus vannamei*) [18,19], rainbow trout (*Oncorhynchus mykiss*) [20,21], red drum (*Sciaenops ocellatus*) [22], silver seabream (*Rhabdosargus sarba*) [23], sobaity sea bream (*Sparidentex hasta*) [24], and young eels (*Anguilla Anguilla*) [25], and suggested that partially substituting FM with PBPM had no negative effects on growth performance and feed utilization [26,27]. Replacing FM with PBPM improved the sensory quality of fillets [28]. However, the appropriate substitution level varied among different studies, and the discrepancy was probably due to many reasons, such as the manufacturing processes and the quality of raw material. Furthermore, supplementation of PBPM together with other animal and plant protein ingredients such as fish protein hydrolysates, hydrolyzed feather meal, and fermented soybean meal was shown to be beneficial to the growth and disease resistance of fish [29,30,31,32].

Coho salmon (*Oncorhynchus kisutch*), an anadromous migratory species, is widely distributed in the North Pacific Ocean, and has become one of the cold-water fish species with the most potential in China. It is famous for its high contents of unsaturated fatty acids (HUFAs) and protein, which has effects in preventing cardiac–cerebral vascular disease and diabetes in human beings [33]. In traditional aquaculture, fish meal and fish oil are the main components of farming salmonids feed; however, with the development of the global aquaculture industry, the shortage resources, the rising price, and other factors, it is necessary to develop diets with low or FM-free products. Until now, studies on protein source substitution in coho salmon feed mainly focused on single-cell protein and soybean as substitutes for FM [34,35], and little information is available on evaluating the effects of replacing FM with animal proteins including PBPM. Thus, the effects of PBPM instead of FM on growth, muscle composition, and tissue biochemical indexes of coho salmon were studied to provide nutritive data for developing cost-effective and environmentally friendly feed.

## 2. Materials and Methods

### 2.1. Experimental Ingredients and Diets

The proximate composition, essential amino acid (EAA) profile, and pepsin in vitro digestibility of FM and PBPM are shown in Table 1. FM, PBPM, and other feed ingredients were provided by Shandong Conqueren Marine Technology Co. Ltd. (Weifang, China). Pepsin in vitro digestibility of FM and PBPM was assayed according to the AOAC [36] method, but slightly modified (pepsin concentration was changed from 0.2% to 0.02%).

Five isonitrogenous (7.45% nitrogen) and isoenergetic (18.61 MJ/kg gross energy) experimental diets were prepared by substituting 0%, 10%, 20%, 40%, and 60% FM protein with the same proportion of PBPM protein, and were designated as PBPM0 (the control), PBPM10, PBPM20, PBPM40, and PBPM60, respectively (Table 2). The protein sources were FM, PBPM, Antarctic krill meal, soybean meal, and corn gluten meal, while the fat sources were soybean oil and fish oil, and the carbohydrate sources were α-starch and high-gluten wheat flour. The dietary EAA profile is shown in Table 3. The solid ingredients were ground into 198 μm powder, mixed well with the oil, and extruded, and the diet pellets were dried at low temperature and packed separately and stored at −20 °C until use.

### 2.2. Fish and Feeding Management

The post-smolts were provided by Conqueren Leading Fresh (Shandong) Marine Science and Technology Inc., Ltd. (Xiashan, Weifang, China) and reared at one of the hatcheries at this company. The control diet was fed to the fish before the formal experimental period. After fasting for 24 h, 10 individuals (each weight: 180.13 ± 1.32 g) were assigned to one of the 15 cages (water capacity 1000 L/cage) with three cages per diet. All the cages were arranged in an earthen pond supplied with filtered underground cold spring water. During the 84-day feeding trial, the post-smolts were manually fed to satiation three times a day (6:50, 11:50, and 16:50). Surplus feed was then collected and dried at 105 °C to obtain the dry weight. The dissolved oxygen, pH value, and water temperature were maintained at 9.5 ± 0.8 mg/L, 6.9 ± 0.3, and 15.5 ± 0.5 °C, respectively.

### 2.3. Sample Collections

After fasting for 24 h, the post-smolts in each cage were anesthetized with tricaine methanesulfonate (MS-222, 30 mg/L), weighed, and counted to measure growth and survival rate. Three post-smolts were sampled from each cage to determine the condition factor (CF), hepatosomatic index (HSI), and viscerosomatic index (VSI). Another five post-smolts were used for collecting serum samples from the caudal veins, and the samples were stored at room temperature for two hours. The serum samples were collected after centrifuging at 3500× *g* for 10 min at 4 °C, and then stored at −80 °C for determination of biochemical parameters. Subsequently, the liver and muscle samples were removed from these five fish and stored at −80 °C for analyses of antioxidative parameters and muscle composition, respectively.

### 2.4. Analytical Methods

#### 2.4.1. Composition Analysis

Proximate content was measured by standard methods of AOAC [36]. Briefly, the content of dry matter was determined by drying at 105 °C to a constant weight. The contents of crude protein, crude lipid, and ash were tested by determining nitrogen (N × 6.25) using the Kjeldahl method, ether extraction using the Soxhlet method, and by heating at 550 °C for 24 h in a muffle furnace, respectively. The test samples were hydrolyzed with 6 mol/L HCl solution at 110 °C for 24 h, and EAAs were measured with automatic amino acid analyzer (Model A300, MembraPure GmbH, German). Gross energy was measured with a Parr 1281 automated oxygen bombardment meter (Parr, Moline, IL, USA).

#### 2.4.2. Biochemical Tissue Analysis

The commercial reagent kits (Nanjing Jiancheng Bioengineering Institute, Nanjing, China) were used to test biochemical tissue parameters. The alanine aminotransferase (ALT) and aspartate aminotransferase (AST) activities of serum samples were tested by the method of Reitman and Frankel [37]. Total cholesterol (TC) and triglyceride (TG) contents were analyzed according to Hardisari and Koiriyah [38]. The serum total protein (TP) level was analyzed according to Grant et al. [39]. High-density lipoprotein (HDL) and low-density lipoprotein (LDL) contents were measured by the production of H_2_O_2_, which underwent peroxidase to produce a red-purple pigment and a POD color rendering reaction, respectively. The serum albumin (ALB) contents were determined according to Doumas et al. [40]. The malondialdehyde (MDA) contents in the liver were determined according to Ayhanci et al. [41]. Superoxide dismutase (SOD) activities in the liver were analyzed by the xanthine oxidase method. Catalase (CAT) activities in the liver were determined according to Kosik-Bogacka et al. [42].

### 2.5. Calculation Methods

The relative formulae [43] were calculated as follows:$$\mathrm{Survival}\;\mathrm{rate}\;(\mathrm{SR},\;\%)=100\;\times \;\frac{\mathrm{final}\;\mathrm{fish}\;\mathrm{number}}{\mathrm{initial}\;\mathrm{fish}\;\mathrm{number}}$$
$$\mathrm{Specific}\;\mathrm{growth}\;\mathrm{rate}\;(\mathrm{SGR},\;\%/\mathrm{day})=100\;\times \;\frac{\ln\;\mathrm{final}\;\mathrm{body}\;\mathrm{weight}\;(g)\;-\;\ln\;\mathrm{initial}\;\mathrm{body}\;\mathrm{weight}\;(g)}{\mathrm{days}}$$
$$\mathrm{Feed}\;\mathrm{coefficient}\;\mathrm{ratio}\;(\mathrm{FCR})=\frac{\mathrm{food}\;\mathrm{intake}\;(g)}{\mathrm{final}\;\mathrm{body}\;\mathrm{weight}\;(g)\;-\;\mathrm{initial}\;\mathrm{body}\;\mathrm{weight}\;(g)}$$
$$\mathrm{Protein}\;\mathrm{efficiency}\;\mathrm{ratio}\;(\mathrm{PER},\;\%)=\frac{\mathrm{final}\;\mathrm{body}\;\mathrm{weight}\;(g)\;-\;\mathrm{initial}\;\mathrm{body}\;\mathrm{weight}\;(g)}{\mathrm{food}\;\mathrm{intake}\;(g)\;\times \;\mathrm{food}\;\mathrm{protein}\;\mathrm{content}\;(\%)}$$
$$\mathrm{CF}\;(g/cm^{3})=100\;\times \;\frac{\;\mathrm{body}\;\mathrm{weight}\;(g)}{{\mathrm{body}\;\mathrm{length}\;(\mathrm{cm})}^{3}}$$
$$\mathrm{HSI}\;(\%)=100\;\times \;\frac{\;\mathrm{liver}\;\mathrm{weight}\;(g)}{\mathrm{body}\;\mathrm{weight}\;(g)}$$
$$\mathrm{VSI}\;(\%)=100\;\times \;\frac{\mathrm{visceral}\;\mathrm{mass}\;\mathrm{weight}}{\mathrm{body}\;\mathrm{weight}}$$

### 2.6. Statistics Analysis

All data are given as mean ± standard deviation (SD), and percentage data were arcsine transformed before analysis. The analyses were performed using SPSS version 25.0. The statistical significance was performed using a one-way analysis of variance (ANOVA) followed by Tukey’s test, and the significant difference was set at *p* < 0.05. The optimum level of PBPM substitution for FM was evaluated by using a broken-line model.

## 3. Results

### 3.1. Growth Performance and Diet Utilization

The survival rate (SR) did not differ significantly (*p* > 0.05) among the dietary groups (Table 4). FBW (final body weight), specific growth rate (SGR), and protein efficiency ratio (PER) all reached the highest values in the PBPM0 (control) group, which was not significantly different (*p* > 0.05) from the PBPM10 and PBPM20 groups, but they showed significant differences (*p* < 0.05) compared with PBPM40 and PBPM60 groups. FCR varied inversely with SGR. The morphological indices (CF, HSI, and VSI) were not significantly affected by the PBPM substitution for FM. Based on broken-line regression analysis of SGR and PER, the optimum dietary PBPM replacing level was evaluated to be 16.63% (Figure 1) and 17.50% (Figure 2) of FM protein for coho salmon post-smolts, respectively (Figure 1).

### 3.2. Muscle Composition and EAA Profile

The dietary PBPM inclusion had no significant (*p* > 0.05) effects on the contents of moisture, crude fat, crude protein, and ash in fish muscle (Table 5). No significant differences were found in muscle contents of leucine, lysine, and methionine between the control and PBPM10 groups, which were significantly higher than those of the other three groups.

### 3.3. Serum Biochemical Parameters

The PBPM60 group had the highest serum ALT and AST activities, which was not significantly (*p* > 0.05) different from the PBPM40 group, but significantly (*p* < 0.05) higher than the control, PBPM10, and PBPM20 groups (Table 6). The control group had significantly higher ALB and TC contents than the other four groups, and the PBPM60 group led to the lowest contents. The control group had the highest TG content, which was not statistically different from the PBPM10 and PBPM20 groups, but significantly higher than the PBPM40 and PBPM60 groups. The HDL, LDL, and TP contents all reached the lowest values in the PBPM60 group and were not different significantly (*p* > 0.05) from the PBPM20 and PBPM40 groups, but they were significantly (*p* < 0.05) lower than the control and PBPM10 groups.

### 3.4. Hepatic MDA Content and Liver Anti-oxidative Enzyme Activity

No significant (*p* > 0.05) differences were observed in the hepatic MDA contents among the PBPM0, PBPM10, and PBPM20 groups, but they increased significantly (*p* < 0.05) with further increase in PBPM replacement level (Table 7). The highest SOD activity was observed in the control group, which did not differ from the PBPM10 group but was significantly higher than the PBPM20, PBPM40, and PBPM60 groups. The CAT activities of the control, PBPM10, and PBPM20 groups were significantly higher than those of the PBPM40 and PBPM60 groups.

## 4. Discussion

The present study showed that partial PBPM substitution for FM had no negative effect on the SR, indicating that the post-smolts had the characteristics of tolerance to the dietary PBPM inclusion. No significant differences were found in SGR and PER among the control, PBPM10, and PBPM20 groups; these values were significantly decreased with further increase in PBPM level, revealing that a high level of dietary PBPM replacement had negative effects on the growth and PER. Using a broken-line model to analyze the relationship between PBPM protein replacement levels and SGR or PER showed that the optimal substitution level of PBPM was 16.63–17.50% of FM protein. FCR showed an inverse trend with SGR and PER. Ma et al. [44] found a similar result that higher than 40% PBPM replacement level significantly decreased the SGR and increased the FCR of golden pompano. However, groups of 15%, 25%, and 35% PBPM protein replacement led to higher final body weight and SGR of juvenile sobaity sea bream than in groups of the control, 45%, and 55% PBPM replacements [45]. Yang et al. [46] found that the SGR and PER were all higher in the PBPM inclusion (40.5–100% of FM protein) groups than those in the control group. Other research has found that PBMP partial replacement of FM has no effect on growth and feed utilization, for example, rainbow trout (PBPM substitution < 66%) [47], snakehead (Channa striata) fingerlings (PBPM substitution < 40%) [48], hybrid grouper (*Epinephelus lanceolatus ♂ × E. fuscoguttatus ♀*, PBPM substitution 40–60%) [49], red porgy (Pagrus pagrus, PBPM substitution 0-70%) [50], and humpback grouper (*Cromileptes altivelis*, PBPM substitution < 100%) [17]. Of course, it also has been found that PBPM can fully replace FM, but some conditions are required, such as required supplementation of EAAs [51,52]. PBPM is the waste of poultry production and processing plants, which quality depends to a large extent on the composition of the raw material and the processing process, including heating, extraction of water, and separation of fat, as well as the time and temperature of the cooking process [53,54,55]. Nile tilapia (*Oreochromis niloticus*) fed diets replacing FM with 10% and 20% fermented PBPM had significantly higher SGR and lower FCR than those fed the control and 40% PBPM groups, which might be explained by the fact that the fermented PBPM enhanced utilization in comparison to the respective untreated fish [56]. Except PBPM containing less certain EAAs and having lower digestibility than fish meal, other factors such as differences in fish species with different metabolism due to different osmotic regulation mechanism, growing stage, experimental period, and different experimental conditions might also influence the results.

The CF, HSI, and VSI contents of the post-smolts were not affected significantly by the PBPM replacement. However, the groups of 10% and 20% PBPM replacements led to remarkably higher HSI than other groups, while the VSI was not affected by the dietary PBPM substitution for yellow catfish (*Pelteobagrus fulvidraco*) [57]. Similar results were found in rainbow trout [58] and Japanese seabass (*Lateolabrax japonicus*) [59]. Kim and Lall [60] suggested that the morphological indices could be affected by dietary macro-nutrients such as protein, lipid, and carbohydrate. Although the feed was formulated as isonitrogenous and isoenergetic, the differences in ingredient digestibility and composition might have effects on the morphological indices.

Generally, the quality of fish fillets can be directly reflected by the fish body composition [61]. Proximate body composition affects several characteristics of fish biology, which have important implications for the development of feeds for farmed fish. PBPM instead of FM as a protein source had no significant effects on muscle proximate composition of coho salmon. Similar results were reported in studies for grouper [48], *Oreochromis niloticus* [62], *Pagrus pagrus* [49], *Sparus aurata* [63], *Tinca tinca* [64], *Lutjanus guttatus* [65], *Sparidentex hasta* [11], *Chanos chanos* [66], yellow catfish [57], Siberian sturgeon (*Acipenser baerii*) [67], and great sturgeon (*Huso huso*) [68]. In comparison, Amirkolaie et al. [46] reported that dry matter and fat content of *Oncorhynchus mykiss* fillets were reduced in the PBPM 100% replacement group, while water content increased with increasing levels of PBPM replacement. High content of EAAs in muscle reflects high nutritional value [69]. In the present study, the contents of leucine, lysine, and methionine in muscles of coho salmon between the control and PBPM10 groups were significantly higher than those of groups with over 20% PBPM protein inclusion, which was in accordance with their contents in the diets. Although the EAA profiles were similar in the diets, Irm et al. [70] found that 30% PBPM replacement of FM resulted in higher lysine and methionine contents in muscles of juvenile black sea bream (*Acanthoparus schlegelii*) than groups of 0, 10%, 20%, 40%, and 60% PBPM replacements, and the latter two groups led to significantly reduced levels. These differences might be influenced by the differences in composition and processing of protein sources, fish species and age, experimental period, and other conditions. The variation in alpha-cellulose and ash contents between the diets also affected the experimental results.

The metabolic function, physiology, and health status are inextricably linked to the dietary nutrition level, which is reflected by the serum biochemical indicators of fish [71]. Energy is produced by TG in serum and stored in fish [72,73]. Cell membrane formation, bile acids, vitamin D, and hormone synthesis all require TC as a source [74]. The contents of TC and TG can reflect the absorption of lipids, while the HDL and LDL represent the decomposition and transport of lipids. The post-smolts fed the control diet exhibited higher serum contents of TC, TG, TP, HDL, and LDL than fish fed with diets including PBPM, which was in accordance with the findings in *Pagrus major* fed a diet blended with fish hydrolysate and animal protein [43], pompano (*Trachinotus blochii*) fed diets with high PBPM [75], and in grouper [48]. These findings indicated that increasing PBPM content in the diet might reduce plasma nutrient content to some extent and further suppress the growth rate. The serum ALB level decreased with increasing dietary PBPM inclusion, suggesting that dietary PBPM inclusion had negative effects on liver function to synthesize protein. ALT and AST are usually used as indicators of liver damage to determine liver health, and higher levels of these enzymes may indicate hepatocyte damage or deleterious effects of feeding regimes [76]. In the present study, both serum ALT and AST contents increased with increasing PBPM inclusion, revealing that PBPM is inferior to FM for the liver health of coho salmon.

MDA is usually used as one of the indexes to assess the extent of oxidative stress [77], and a high MDA level can induce programmed cell death [78]. An appropriate amount of dietary PBPM inclusion level (substituting < 40% FM) did not affect the antioxidant capacity of post-smolts, but the added amount should not be too high. CAT and SOD are the main antioxidant enzymes that protect cells and tissues from oxidative damage by producing ROS and catalyzing the production of hydrogen peroxide (H_2_O_2_) to produce oxygen and water [79,80,81]. Compared with the control group, the low content of dietary PBPM had no significant effects on the activities of SOD and CAT in the liver of coho salmon, which were significantly decreased in the group fed high amounts of PBPM. Zhou et al. [82] found that the SOD and CAT activities in the liver of juvenile cobia (*Rachycentron canadum*) were not significantly affected by the replacement of FM with PBPM. The effects of PBPM instead of FM on antioxidant enzymes, immune system, and health status are worthy of further study.

## 5. Conclusions

In conclusion, the growth performance, feed utilization, muscle composition, serum biochemical indices, and liver antioxidant enzyme activities of coho salmon were negatively affected by high dietary inclusion level of PBPM. Higher PBPM (substituting *<* 40% FM) will affect liver antioxidant capacity and health. Based on SGR and PER, the optimal substitution level of PBPM protein was 16.63–17.50% of FM protein for coho salmon post-smolts using broken-line regression analysis.

## Figures and Tables

**Figure 1 animals-13-02789-f001:**
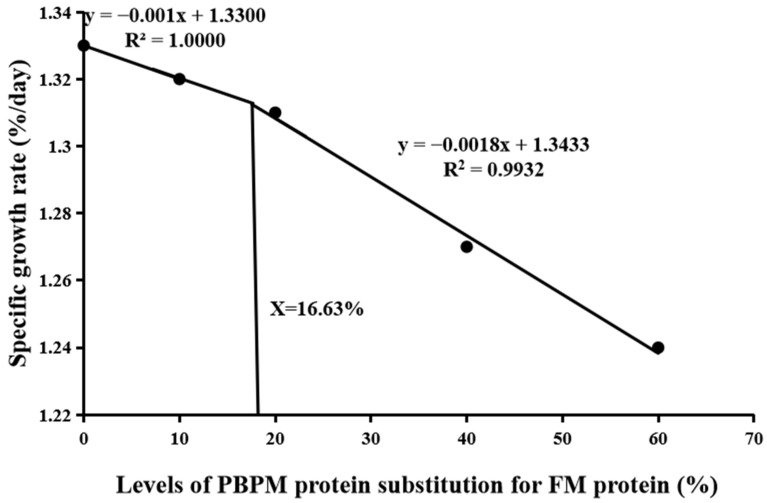
A broken-line regression analysis of the relationship between specific growth rate (SGR) and PBPM inclusion level evaluated that the optimal substitution level of PBPM protein was 16.63% of FM protein for coho salmon (*Oncorhynchus kisutch*) post-smolts.

**Figure 2 animals-13-02789-f002:**
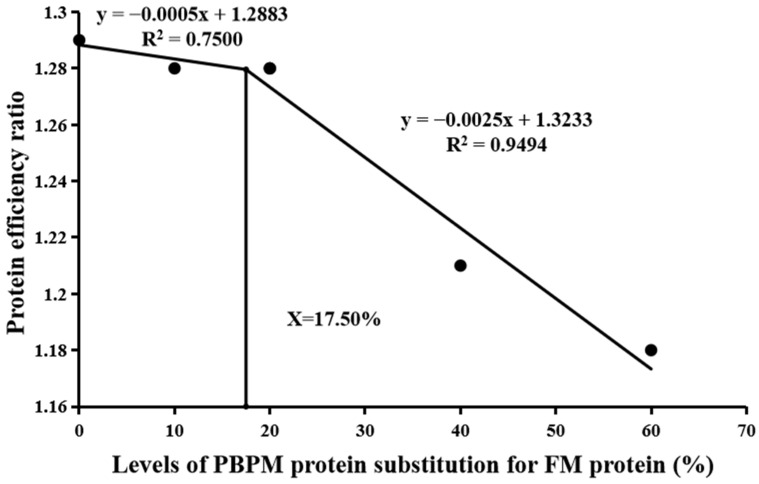
A broken-line regression analysis of the relationship between protein efficiency ratio (PER) and PBPM inclusion level evaluated that the optimal substitution level of PBPM protein was 17.50% of FM protein for coho salmon (*Oncorhynchus kisutch*) post-smolts.

**Table 1 animals-13-02789-t001:** Proximate composition, EAA profile, and pepsin in vitro protein digestibility of fish meal and poultry by-product meal (% dry matter)**.**

Ingredients	Fish Meal	Poultry by-Product Meal
Proximate composition and pepsin in vitro protein digestibility		
Moisture (%)	8.27	5.52
Crude protein (%)	70.21	66.52
Crude lipid (%)	11.55	13.63
Ash (%)	17.08	12.57
Gross energy (MJ/kg)	21.80	22.87
Pepsin in vitro digestibility (%)	84.03	92.41
EAA profile (g/kg crude protein)		
Arginine	64.73	56.28
Histidine	26.79	23.73
Isoleucine	50.84	41.97
Leucine	70.43	57.12
Lysine	80.96	63.72
Methionine	29.94	23.97
Phenylalanine	53.76	42.39
Threonine	39.85	36.51
Valine	51.03	45.45
Cystine	8.30	9.66
Tyrosine	40.52	37.96

Abbreviations: EAA, essential amino acid.

**Table 2 animals-13-02789-t002:** Formulation and proximate composition of the experimental diets for coho salmon (*Oncorhynchus kisutch*) post-smolts (% dry matter)**.**

Ingredients	Diets (Designated Percentage of PBPM Replacement Levels)				
	PBPM0 (Control)	PBPM10	PBPM20	PBPM40	PBPM60
Peru fish meal	40.00	36.00	32.00	24.00	16.00
Poultry by-product meal	0.00	4.20	8.40	16.80	25.20
Antarctic krill meal ^1^	7.00	7.00	7.00	7.00	7.00
Soybean meal ^1^	15.00	15.00	15.00	15.00	15.00
Corn gluten meal ^1^	6.00	6.00	6.00	6.00	6.00
High-gluten wheat flour ^1^	16.25	16.25	16.25	16.25	16.25
α-starch ^1^	2.50	2.50	2.50	2.50	2.50
Soybean lecithin ^1^	1.00	1.00	1.00	1.00	1.00
Soybean oil ^1^	4.00	3.92	3.84	3.68	3.52
Fish oil ^1^	4.00	4.00	4.00	4.00	4.00
Mono-calcium phosphate ^1^	1.00	1.00	1.00	1.00	1.00
Mineral premix ^2^	1.00	1.00	1.00	1.00	1.00
Vitamin premix ^3^	1.00	1.00	1.00	1.00	1.00
Choline chloride	0.40	0.40	0.40	0.40	0.40
Ascorbic acid phosphate (35%)	0.10	0.10	0.10	0.10	0.10
α-cellulose	0.72	0.60	0.48	0.24	0.00
Ethoxyquin (60%)	0.03	0.03	0.03	0.03	0.03
Proximate composition					
Dry matter	89.14	90.04	88.95	91.23	90.35
Crude protein	46.95	46.76	46.41	46.10	46.43
Crude lipid	15.15	15.55	15.25	15.36	15.75
Ash	9.81	9.66	9.47	8.93	8.56
Gross energy (MJ/kg)	18.71	18.77	18.63	18.47	18.49

^1^ Provided by Shandong Conqueren Marine Technology Co., Ltd., Weifang, China. ^2^ Composition (g/kg mineral premix): AlK(SO_4_)_2_⋅12H_2_O, 123.7; CuSO_4_⋅5H_2_O, 32.0; CoCl_2_⋅6H_2_O, 49.0; FeSO_4_⋅7H_2_O, 707.0; MgSO_4_⋅7H_2_O, 4317.0; MnSO_4_⋅4H_2_O, 31.0; KI, 5.3; NaCl, 4934.0; Na_2_SeO_3_⋅H_2_O, 3.4; ZnSO_4_⋅7H_2_O, 177.0. ^3^ Composition (IU or g/kg vitamin premix): retinal palmitate, 10,000 IU; cholecalciferol, 4000 IU; α-tocopherol, 75.0 IU; menadione, 22.0 g/kg; thiamine–HCl, 40.0 g/kg; riboflavin, 30.0 g/kg; D-calcium pantothenate, 150.0 g/kg; pyridoxine–HCl, 20.0 g/kg; meso-inositol, 500.0 g/kg; D-biotin, 1.0 g/kg; folic acid, 15.0 g/kg; niacin, 300.0 g/kg; cyanocobalamin, 0.3 g/kg.

**Table 3 animals-13-02789-t003:** EAAs profile (g/kg crude protein) of the experimental diets for coho salmon (*Oncorhynchus kisutch*) post-smolts.

EAAs ^1^	Diets (Designated Percentage of PBPM Replacement Levels)				
	PBPM0 (Control)	PBPM10	PBPM20	PBPM40	PBPM60
Arginine	62.27	62.05	61.82	61.46	61.08
Histidine	24.07	23.98	23.85	23.63	23.45
Isoleucine	49.72	49.60	49.45	49.07	48.69
Leucine	82.66	82.43	82.17	81.79	81.36
Lysine	68.92	68.51	68.25	67.56	67.03
Methionine	28.70	28.48	28.17	27.87	27.51
Phenylalanine	54.63	54.47	54.25	53.94	53.57
Threonine	46.67	46.54	46.44	46.29	46.02
Valine	50.08	49.91	49.76	49.55	49.26
Cystine	9.44	9.47	9.52	9.68	9.87
Tyrosine	39.33	39.18	38.91	38.87	38.79

^1^ No tryptophan was detected because of acid hydrolysis. Abbreviations: EAA, essential amino acid.

**Table 4 animals-13-02789-t004:** Growth performance and feed utilization of coho salmon (Oncorhynchus kisutch) post-smolts fed diets partially substituting FM with PBPM for 12 weeks.

Parameters	Diets (Designated Percentage of PBPM Replacement Levels)				
	PBPM0 (Control)	PBPM10	PBPM20	PBPM40	PBPM60
SR (%)	96.67 ± 0.57	100 ± 0.00	100 ± 0.00	96.67 ± 0.58	96.67 ± 0.58
IBW (g)	180.62 ± 1.47	179.09 ± 0.92	179.73 ± 1.15	180.21 ± 1.87	181.02 ± 1.19
FBW (g)	550.49 ± 5.07 ^b^	545.02 ± 3.51 ^b^	540.15 ± 3.29 ^b^	525.74 ± 2.93 ^a^	514.75 ± 4.73 ^a^
SGR (%/day)	1.33 ± 0.01 ^b^	1.32 ± 0.01 ^b^	1.31 ± 0.01 ^b^	1.27 ± 0.01 ^a^	1.24 ± 0.01 ^a^
FCR	1.56 ± 0.03 ^a^	1.59 ± 0.01 ^a^	1.58 ± 0.01 ^a^	1.78 ± 0.02 ^b^	1.79 ± 0.03 ^b^
PER	1.29 ± 0.02 ^b^	1.28 ± 0.01 ^b^	1.28 ± 0.01 ^b^	1.21 ± 0.03 ^a^	1.18 ± 0.03 ^a^
CF (g/cm^3^)	1.67 ± 0.07	1.68 ± 0.05	1.60 ± 0.04	1.65 ± 0.02	1.64 ± 0.05
HSI (%)	1.01 ± 0.03	1.07 ± 0.04	1.06 ± 0.03	0.97 ± 0.03	0.99 ± 0.03
VSI (%)	7.72 ± 0.23	7.32 ± 0.46	7.30 ± 0.35	7.43 ± 0.53	8.00 ± 0.32

Values are presented as mean ± SD of three replicate groups. Means in the same row with different superscript letters are significantly different (*p* < 0.05). Abbreviations: IBW, initial body weight; FBW, final bodyweight; SGR, specific growth rate; FCR, feed conversion ratio; PER, protein efficiency ratio; CF, condition factor; HSI, hepatosomatic index; VSI, viscerosomatic index.

**Table 5 animals-13-02789-t005:** Muscle proximate composition and EAA profile of coho salmon (*Oncorhynchus kisutch*) post-smolts fed diets partially substituting FM with PBPM for 12 weeks.

Parameters	Diets (Designated Percentage of PBPM Replacement Levels)				
	PBPM0 (Control)	PBPM10	PBPM20	PBPM40	PBPM60
Proximate composition (% wet weight)					
Moisture (%)	73.40 ± 0.37	73.26 ± 0.25	73.39 ± 0.32	73.14 ± 0.21	73.18 ± 0.46
Crude protein (%)	20.69 ± 0.14	20.51 ± 0.28	20.35 ± 0.19	20.63 ± 0.22	20.65 ± 0.26
Crude lipid (%)	3.38 ± 0.15	3.49 ± 0.16	3.46 ± 0.14	3.27 ± 0.16	3.35 ± 0.12
Ash (%)	2.43 ± 0.12	2.35 ± 0.15	2.50 ± 0.15	2.34 ± 0.09	2.26 ± 0.10
EAAs profile (g/kg crude protein)					
Arginine	74.41 ± 0.39	75.59 ± 0.23	73.68 ± 0.25	73.96 ± 0.30	73.73 ± 0.44
Histidine	28.64 ± 0.22	28.21 ± 0.17	27.69 ± 0.31	28.25 ± 0.28	28.07 ± 0.26
Isoleucine	58.48 ± 0.35	58.22 ± 0.29	57.97 ± 0.21	57.83 ± 0.23	56.97 ± 0.16
Leucine	97.53 ± 0.37 ^b^	96.67 ± 0.42 ^ab^	96.13 ± 0.34 ^a^	95.75 ± 0.25 ^a^	95.11 ± 0.28 ^a^
Lysine	82.65 ± 0.26 ^b^	82.22 ± 0.33 ^ab^	81.77 ± 0.52 ^a^	81.39 ± 0.65 ^a^	80.55 ± 0.22 ^a^
Methionine	34.18 ± 0.25 ^b^	33.12 ± 0.34 ^ab^	32.61 ± 0.27 ^a^	32.06 ± 0.36 ^a^	31.97 ± 0.20 ^a^
Phenylalanine	65.46 ± 0.31	65.69 ± 0.23	64.80 ± 0.46	64.55 ± 0.24	64.21 ± 0.53
Threonine	55.52 ± 0.29	55.16 ± 0.22	54.63 ± 0.25	54.51 ± 0.54	56.66 ± 0.33
Valine	59.65 ± 0.23	60.07 ± 0.35	59.26 ± 0.36	58.67 ± 0.39	57.19 ± 0.47
Cystine	12.23 ± 0.15	11.73 ± 0.21	12.36 ± 0.27	12.91 ± 0.10	12.84 ± 0.12
Tyrosine	45.90 ± 0.39	45.54 ± 0.37	45.29 ± 0.33	45.89 ± 0.18	45.94 ± 0.30

Values are presented as mean ± SD of three replicate groups. Means in the same row with different superscript letters are significantly different (*p* < 0.05). Abbreviations: EAA, essential amino acid.

**Table 6 animals-13-02789-t006:** Serum biochemical parameters of coho salmon (*Oncorhynchus kisutch*) post-smolts fed diets partially substituting FM with PBPM for 12 weeks.

Parameters	Diets (Designated Percentage of PBPM Replacement Levels)				
	PBPM0 (Control)	PBPM10	PBPM20	PBPM40	PBPM60
ALB mg/mL	26.15 ± 0.79 ^c^	22.27 ± 0.89 ^b^	21.79 ± 1.01 ^ab^	20.98 ± 0.69 ^ab^	18.70 ± 0.93 ^a^
ALT U/mL	6.37 ± 0.31 ^a^	6.59 ± 0.28 ^a^	6.83 ± 0.24 ^a^	7.96 ± 0.27 ^b^	8.51 ± 0.19 ^b^
AST U/mL	5.42 ± 0.34 ^a^	6.45 ± 0.27 ^ab^	7.27 ± 0.28 ^b^	8.74 ± 0.33 ^c^	9.80 ± 0.36 ^c^
HDL nmol/mL	0.37 ± 0.02 ^b^	0.35 ± 0.02 ^b^	0.32 ± 0.02 ^ab^	0.29 ± 0.02 ^ab^	0.26 ± 0.01 ^a^
LDL nmol/mL	0.93 ± 0.04 ^b^	0.87 ± 0.03 ^b^	0.81 ± 0.04 ^ab^	0.75 ± 0.03 ^ab^	0.70 ± 0.03 ^a^
TC nmol/mL	8.54 ± 0.28 ^c^	7.35 ± 0.24 ^b^	7.29 ± 0.19 ^b^	6.22 ± 0.33 ^a^	5.84 ± 0.29 ^a^
TG nmol/mL	1.90 ± 0.11 ^b^	1.75 ± 0.07 ^ab^	1.72 ± 0.12 ^ab^	1.66 ± 0.05 ^a^	1.60 ± 0.08 ^a^
TP nmol/mL	48.59 ± 0.53 ^b^	47.97 ± 0.38 ^b^	46.87 ± 0.40 ^ab^	45.39 ± 0.45 ^ab^	43.71 ± 0.36 ^a^

Values are presented as mean ± SD of three replicate groups. Means in the same row with different superscript letters are significantly different (*p* < 0.05). Abbreviations: ALB, albumin; ALT, alanine aminotransferase; AST, aspartate aminotransferase; HDL, high-density lipoprotein; LDL, low-density lipoprotein; TC, total cholesterol; 6; TP, total protein.

**Table 7 animals-13-02789-t007:** Liver MDA and anti-oxidative enzyme activities of coho salmon (*Oncorhynchus kisutch*) post-smolts fed diets partially substituting FM with PBPM for 12 weeks.

Parameters	Diets (Designated Percentage of PBPM Replacement Levels)				
	PBPM0 (Control)	PBPM10	PBPM20	PBPM40	PBPM60
MDA (nmol/mg)	8.22 ± 0.86 ^a^	9.68 ± 1.02 ^a^	9.79 ± 0.87 ^a^	12.56 ± 0.83 ^b^	15.10 ± 1.42 ^c^
SOD (U/mg)	209.55 ± 22.33 ^b^	163.18 ± 29.83 ^ab^	150.57 ± 21.22 ^a^	131.20± 19.25 ^a^	117.91 ± 25.33 ^a^
CAT (U/mg)	35.77 ± 1.33 ^b^	33.94 ± 2.24 ^b^	30.56 ± 1.56 ^b^	23.23 ± 1.03 ^a^	21.53 ± 0.66 ^a^

Values are presented as mean ± SD of three replicate groups. Means in the same row with different superscript letters are significantly different (*p* < 0.05). Abbreviations: MDA, Malondialdehyde; SOD, superoxide dismutase; CAT, catalase.

## Data Availability

The data that support the findings of this study are available from the corresponding author upon reasonable request.

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
