# Peer review of "Partial Replacement of Fishmeal with Poultry By-Product Meal in Diets for Coho Salmon (Oncorhynchus kisutch) Post-Smolts"

_animals, 2023, doi:10.3390/ani13172789_

Round 1
Reviewer 1 Report
Comments and suggestions: Partial replacement of fishmeal with poultry by-product meal in diets for coho salmon (Oncorhynchus kisutch) post-smolts
This paper is interesting and the experiments are organized. However, there are some grammar/spelling and clarification points that I have suggested to improve the text, both to clarify and for a better flow of the text. The grammar/spelling suggestions apply to more than the lines indicated here.
· Lines 11-16: Correspondence should be shorter.
· Lines 17,31, etc.: Fish meal should be one word, fishmeal, but is however different in the text – lines 17, 31, etc.
· Lines 23-24: Spelling: in comparison to (not with)
· Lines 24: “Few information is available” is wrong grammar, few is in combination with plural countable nouns. Exchange for “little” or change the sentence.
· Lines 87-89: Underlined has to be rephrased.
o “It is very popular with consumers due to it’s rich in a variety of highly unsaturated fatty acids, which can effectively guard against a variety of cardiovascular and cerebrovascular diseases and diabetes”.
· Lines 89-90: Rephrase the first half of the following sentence and explain better the second half of it (free fish products?!”)
o “Farming salmonids are traditionally used FM and fish oil as the main ingredients which require to develop diets with low or free fish products.”
· Table 1: Fix the alignment of the columns. In total, the percentage of fishmeal is 107% for the ingredients and 98% for the PBPB. Furthermore, results should be shown with standard deviation.
· Table 1: EAA profile for fishmeal is in total <520 g/kg crude protein. Where is the rest? Consider adding “other” to account for the rest, as the EAA profile includes the whole profile as nothing else is indicated.
· Line 99-103: Why is Pepsin measured specifically? Please clarify that in the text.
· Line 116: Table 2: The manuscript does not mention alpha-cellulose although it is listed in Table 2 and varies between the diets. This needs to be addressed and discussed if it can affect the results.
· Line 116: Table 2: Crude lipids and ash also vary in different diets. Discuss if and how it can affect the results.
· Table 3: There is 62.27 for the control in PBPM which was 64.73 in Table 1. Why? Standard deviation should be included in all tables.
· Lines 174-182: I would have liked to see references for the formulas, so it would be easy to spot that these are known formulas used in academia.
· Line 191: I recommend stating the SR abbreviation if it is in a new chapter and has not been mentioned in text before (just displayed in a formula). Same comment for the other abbreviations in line 192.
· Line 201: Table 4: Here is the survival rate not in abbreviation but in all the other parameters. Please keep consistent.
· Line 192: FBW of the control group is not the highest, the Tukey’s test you did shows marked with small numbers that PBPM0, PBPM10, and PBPM20 are not significantly different. Therefore, you cannot say that so please rephrase the sentence.
· Line 206: A clarification of why broken-line regression is performed to get the optimal feed is needed.
· Lines 230-232: “The lowest HDL, LDL, and TP contents were observed in the PBPM60 group, which were not different significantly from PBPM20 and PBPM40 groups, but significantly lower than the control and PBPM10 group.” Consider rephrasing – same comment as in line 192.
· Lines 236-238: Consider having this in smaller letters and with less line space.
· Line: 244: Consider exchanging “remarkably” for significantly.
· Lines 267-286: Consider rephrasing this paragraph to more of a discussion as it reads rather like a list of who found what, needs some discussion around it, or perhaps to sum it together. Furthermore, discuss why some species can be replaced with PBPM and some not.
· Line 286: There is nothing that is called waste, but rather as side streams. Moreover, you have clarified what PBPM is, so this sentence is out of place here.
· Line 346-348: “The low dietary PBPM inclusion level did not significantly affect the MDA level in liver of coho salmon. However, the groups of PBPM40 and PBPM60 led to significantly increased […]” This must be clearer.
· Line 360-362: Liver health was affected by higher PBPM (40 and 60), which I would mention in the conclusion. MDA, CAT, and SOD are not mentioned in the conclusions.
Author Response
Response to Reviewer 1 Comments
Point 1: Lines 11-16: Correspondence should be shorter.
Response 1: Thank you for your suggestions!
We have removed the corresponding author address to make correspondence shorter.
Point 2: Lines 17,31, etc.: Fish meal should be one word, fishmeal, but is however different in the text – lines 17, 31, etc.
Response 2: We added the spaces between “fish” and “meal”.
Point 3: Lines 23-24: Spelling: in comparison to (not with)
Response 3: Sorry, we have revised.
Point 4: Lines 24:“Few information is available” is wrong grammar, few is in combination with plural countable nouns. Exchange for “little” or change the sentence.
Response 4: We have revised “few” to” little”.
Point 5: Lines 87-89: Underlined has to be rephrased.
o “It is very popular with consumers due to it’s rich in a variety of highly unsaturated fatty acids, which can effectively guard against a variety of cardiovascular and cerebrovascular diseases and diabetes”.
Response 5: We have revised to “It is famous for its highly contents of unsaturated fatty acids (HUFAs)and protein, which had effect in preventing cardiac-cerebral vascular disease and diabetes for human being”.
Point 6:Lines 89-90: Rephrase the first half of the following sentence and explain better the second half of it (free fish products?!”)
o“Farming salmonids are traditionally used FM and fish oil as the main ingredients which require to develop diets with low or free fish products.”
Response 6: We have revised to “In traditional aquaculture, fish meal and fish oil are the main components of farming salmonids feed, however, with the development of the global aquaculture industry, the shortage resources, the rising price and other factors, which require to develop diets with low or FM-free products.”
Point 7: Table 1: Fix the alignment of the columns. In total, the percentage of fishmeal is 107% for the ingredients and 98% for the PBPB. Furthermore, results should be shown with standard deviation.
Response 7: The contents of crude protein, crude lipid and ash were percentages in the dry matter, which did not include moisture content.
Point 8: Table 1: EAA profile for fishmeal is in total < 520 g/kg crude protein. Where is the rest? Consider adding “other” to account for the rest, as the EAA profile includes the whole profile as nothing else is indicated.
Response 8: The EAA differences are usually compared between the ingredients.
Point 9: Line 99-103: Why is Pepsin measured specifically? Please clarify that in the text.
Response 9: The pepsin in vitro digestibility of FM and PBPM was assayed according to AOAC slightly modified method (pepsin concentration was change from 0. 2% to 0.02%), which is described in the text.
Point 10: Line 116: Table 2: The manuscript does not mention alpha-cellulose although it is listed in Table 2 and varies between the diets. This needs to be addressed and discussed if it can affect the results.
Response 10: According to Reviewer’s suggestion, we have revised in the discussion.
Point 11: Line 116: Table 2: Crude lipids and ash also vary in different diets. Discuss if and how it can affect the results.
Response 11: The ash varied in different diets. According to Reviewer’s suggestion, we have revised in the discussion.
Point 12: Table 3: There is 62.27 for the control in PBPM which was 64.73 in Table 1. Why? Standard deviation should be included in all tables.
Response 12: 62.27 is the arginine percentage in the control diet in Table 3, while 64.73 is the arginine percentage in the ingredient of PBPM in Table 1.
Point 13: Lines 174-182: I would have liked to see references for the formulas, so it would be easy to spot that these are known formulas used in academia.
Response 13: yes, we added references.
Point 14: Line 191: I recommend stating the SR abbreviation if it is in a new chapter and has not been mentioned in text before (just displayed in a formula). Same comment for the other abbreviations in line 192.
Response 14: yes, we have added comment for abbreviations in line 191 and 192.
Point 15: Line 201: Table 4: Here is the survival rate not in abbreviation but in all the other parameters. Please keep consistent.
Response 15: We have revised to”SR”.
Point 16: Line 192: FBW of the control group is not the highest, the Tukey’s test you did shows marked with small numbers that PBPM0, PBPM10, and PBPM20 are not significantly different. Therefore, you cannot say that so please rephrase the sentence.
Response 16: We have revised to ”FBW, SGR and PER all reached the highest values in PBPM0 (control) group, which was not different significantly (p > .05) from the PBPM10 and PBPM20 groups, but they showed significant differences (p < .05) compare with PBPM40 and PBPM60 groups”.
Point 17: Line 206: A clarification of why broken-line regression is performed to get the optimal feed is needed.
Response 17: After comparing the different regression models, the broken-line regression is the most suitable, in which the R2 is the highest.
Point 18: Lines 230-232: “The lowest HDL, LDL, and TP contents were observed in the PBPM60 group, which were not different significantly from PBPM20 and PBPM40 groups, but significantly lower than the control and PBPM10 group.” Consider rephrasing – same comment as in line 192.
Response 18: We have revised to “The HDL, LDL and TP contents all reached the lowest values in PBPM60 group, which were not different significantly (p > .05) from PBPM20 and PBPM40 groups, but they showed significantly (p < .05) lower than the control and PBPM10 groups.”
Point 19: Lines 236-238: Consider having this in smaller letters and with less line space.
Response 19: Thank you, manuscripts have adjusted.
Point 20: Line: 244: Consider exchanging “remarkably” for significantly.
Response 20: The manuscripts have revised.
Point 21: Lines 267-286: Consider rephrasing this paragraph to more of a discussion as it reads rather like a list of who found what, needs some discussion around it, or perhaps to sum it together. Furthermore, discuss why some species can be replaced with PBPM and some not.
Response 21: According to Reviewer’s suggestion, we have revised in the discussion.
Point 22: Line 286: There is nothing that is called waste, but rather as side streams. Moreover, you have clarified what PBPM is, so this sentence is out of place here.
Response 22: We have revised to”PBPM as the waste of poultry production and processing plants, which quality depends to a large extent on the composition of the raw material and the processing process, including heating, extraction of water and separation of fat, as well as the time and temperature of the cooking process”.
Point 23: Line 346-348: “The low dietary PBPM inclusion level did not significantly affect the MDA level in liver of coho salmon. However, the groups of PBPM40 and PBPM60 led to significantly increased […]” This must be clearer.
Response 23: We have revised to” An appropriate amount of dietary PBPM inclusion level (substituting < 40% FM) did not affect on the antioxidant capacity of post-smolts, but the added amount should not be too high”.
Point 24: Line 360-362: Liver health was affected by higher PBPM (40 and 60), which I would mention in the conclusion. MDA, CAT, and SOD are not mentioned in the conclusions.
Response 24: We have added ” Higher PBPM (substituting < 40% FM) will affect liver antioxidant capacity and health”in the conclusions.
Once again, thank you very much for your comments and suggestion.

Reviewer 2 Report
Animals-2481771
Partial replacement of fishmeal with poultry by-product meal in diets
for coho salmon (Oncorhynchus kisutch) post-smolts, by Yu et al.
The conclusion could not be led from the results of this work. The growth and feed utilization parameters as well as serum and liver biochemical parameters decreased/increased as the inclusion level of PBPM increased. The use of broken-line model for the data in this work is unsuitable. In addition, the duration of experiment (84 days) are too short and the final fish sizes are too small relative to a practical aquaculture of this species. So, the conclusion should be that PBPM used in this work is not suitable as a substitute of fishmeal in coho salmon diet.
The authors need to have checked the oxidation of PBPM used in this work, by analyzing, for example, peroxide value, because antioxidative status of fish got worse with the level of dietary PBPM increased. Unlike fishmeal for aquafeed imported from foreign countries, antioxidants such as ethoxyquin and BHT are usually un-supplemented to livestock protein meals including PBPM, so often very high POV (more than 200 meq/kg lipid) are found in these animal protein ingredients. So, feed manufacturers usually quickly spend such ingredients after they obtained fresh ones from renderers. See also the study by Higgs et al. (1979) where they found a good results of “defatted” PBPM inclusion in coho salmon diet, which may support my idea (peroxides might have been removed by defatting).
Higg. D.A. et al. (1979) Development of practical dry diets for coho salmon, Oncorhynchus kisutch, using poultry by-product meal, feather meal, soybean meal and rapeseed meal as major protein sources. In: Proceeding of the World Symposium of Finfish Nutrition and Fish Feed Technology. Hamburg 20-23 June, 1978. Vol.II, 191-218.
Total amount of feed intake (g/fish) could be estimated from the FCR data or PER data presented in Table 4. However, the total feed intake values estimated from FCR (647, 648, 642, 674, 661 g/fish) and PER (611, 611, 607, 619, 609 g/fish) are quite different. Why such large differences happen?
L40 and elsewhere: not “feed coefficient ratio” but correctly “feed conversion ratio”
Tables 1, 3 and 5: Most of cystine were destroyed by the authors’ acid digestion method (L156-L157).
L111: Here the authors state “shrimp powder”, but in Table 2, “krill meal” is listed. Which is correct?
L131-L139: First, the authors need to state the place where the feeding experiment was performed. Next, the authors also need to state how they quantified the amounts of feed pellets leftover.
L145: I cannot understand the statement, “smolts were stored at room temperature for two hours”. Did the authors take serum etc. from the dead fish?
L162-L163: “The homogenated samples of …activities” is not clear.
L187-L188: A broken-line model could not be fit for the SGR data. A regression analysis is better.
L196-L200: Not correct as mentioned above.
L249: Liver “MDA and” anti-oxidative…
L265: sobaity “sea bream” than…
L280: not “sun bass” but “sunshine bass”
Careless mistakes should be corrected, for example, misuse of adverb as if
adjective, tense, lack of definite article.
Author Response
Response to Reviewer 2 Comments
Point 1: The conclusion could not be led from the results of this work. The growth and feed utilization parameters as well as serum and liver biochemical parameters decreased/increased as the inclusion level of PBPM increased. The use of broken-line model for the data in this work is unsuitable. In addition, the duration of experiment (84 days) are too short and the final fish sizes are too small relative to a practical aquaculture of this species. So, the conclusion should be that PBPM used in this work is not suitable as a substitute of fishmeal in coho salmon diet.
Thank you for your suggestions!
Referring to the study of Hassani et al., the method of broken line model is also used to analyze the optimal replacement fish meal with poultry by-products.
Additionally, we noticed most replacement trials were conducted 8-12 weeks, which might be short to coho salmon species, and we would extend the duration in the future feeding trial.
Hassani M.H.S., Banavreh A., Jourdehi A.Y., et al. The feasibility of partial replacement fish meal with poultry by-products in practical diets of juvenile great sturgeon, Huso huso: Effects on growth performance, body composition, physiometabolic indices, digestibility and digestive enzymes[J]. Aquaculture Research, 2021, 52(8): 3605-3616.
Point 2: The authors need to have checked the oxidation of PBPM used in this work, by analyzing, for example, peroxide value, because antioxidative status of fish got worse with the level of dietary PBPM increased. Unlike fishmeal for aquafeed imported from foreign countries, antioxidants such as ethoxyquin and BHT are usually un-supplemented to livestock protein meals including PBPM, so often very high POV (more than 200 meq/kg lipid) are found in these animal protein ingredients. So, feed manufacturers usually quickly spend such ingredients after they obtained fresh ones from renderers. See also the study by Higgs et al. (1979) where they found a good results of “defatted” PBPM inclusion in coho salmon diet, which may support my idea (peroxides might have been removed by defatting).
Higg. D.A. et al. (1979) Development of practical dry diets for coho salmon, Oncorhynchus kisutch, using poultry by-product meal, feather meal, soybean meal and rapeseed meal as major protein sources. In: Proceeding of the World Symposium of Finfish Nutrition and Fish Feed Technology. Hamburg 20-23 June, 1978. Vol.II, 191-218.
In fact, we did not consider the oxidation of PBPM, which would be focused in future research.
Point 3: Total amount of feed intake (g/fish) could be estimated from the FCR data or PER data presented in Table 4. However, the total feed intake values estimated from FCR (647, 648, 642, 674, 661 g/fish) and PER (611, 611, 607, 619, 609 g/fish) are quite different. Why such large differences happen?
We rechecked the data, and found that the total amount of feed intake estimated from PER was correct. The total amount of feed intake estimated from FCR should deduct the moisture from the feed. The FCR was recalculated and showed in Table 4.
Point 4: L40 and elsewhere: not “feed coefficient ratio” but correctly “feed conversion ratio”
We have revised “feed coefficient ratio” to” feed conversion ratio”.
Tables 1, 3 and 5: Most of cystine were destroyed by the authors’ acid digestion method (L156-L157).
Indeed, the acid digestion might destroy cystine, the analyzed content was lower than the initial content.
L111: Here the authors state “shrimp powder”, but in Table 2, “krill meal” is listed. Which is correct?
We have revised to “antarctic krill meal”
L131-L139: First, the authors need to state the place where the feeding experiment was performed. Next, the authors also need to state how they quantified the amounts of feed pellets leftover.
We have revised to”The post-smolts were provided by Conqueren Leading Fresh (Shandong) Marine Science & Technology Inc., Ltd. (Xiashan, Weifang, China) and reared at one of the hatcheries in this company” and added “Feed to apparent satiation, collect surplus feed and then dried at 105°C to gain the dry weight”.
L145: I cannot understand the statement, “smolts were stored at room temperature for two hours”. Did the authors take serum etc. from the dead fish?
We have revised to “Another five post-smolts were used for collect serum samples from the caudal veins, and the samples stored at room temperature for two hours”.
L162-L163: “The homogenated samples of …activities” is not clear.
We have revised to “The alanine aminotransferase (ALT) and aspartate aminotransferase (AST) activities of serum samples were tested by the method of Reitman and Frankel”.
L187-L188: A broken-line model could not be fit for the SGR data. A regression analysis is better.
After comparing the different regression models, the broken-line regression is the most suitable, in which the R2 is the highest.
L196-L200: Not correct as mentioned above.
We have revised to”Based on the broken-line model analysis of SGR and PBPM protein levels replacing with the same portions of FM protein , the optimal substitution level of PBPM protein was 16.63% of FM protein for coho salmon post- smolts”
L249: Liver “MDA and” anti-oxidative…
We have revised to”Liver MDA and anti-oxidative…”.
L265: sobaity “sea bream” than…
We have revised to”sea bream”.
L280: not “sun bass” but “sunshine bass”
We have revised to”sunshine bass”.
Once again, thank you very much for your comments and suggestions.

Reviewer 3 Report
A significant re-organization of the Results and Discussion sections is required (see attached file)

Author Response
Response to Reviewer 3 Comments
1) A brief summary (one short paragraph) outlining the aim of the paper, its main contributions and strengths.
This manuscript is a significant contribution to the subject of fish nutrition, as it has been thoroughly designed and provides lots of experimental data and results.
Thanks a lot for Reviewer’s comments.
2) General concept comments
Article: highlighting areas of weakness, the testability of the hypothesis, methodological inaccuracies, missing controls, etc.
The testability of the hypothesis is sound. The methodology followed and the experimental design are valid and sound as well.
Thanks a lot for Reviewer’s general comments.
Is SGR the ONLY criterion in order to choose the correct level of FM substitution (16.63%), OR should we take into account the rest of indices (i.e. Protein Efficiency Ration and/OR Feed Coefficient Ratio which is inversely related to SGR). Where, at which % of FM subistitution, is the “Golden Ratio”-“fine balance” of all the parameters? Don’t you have to take into account the rest of liver and serum blood indices, in conjunction with SGR, PER, etc?
According to Reviewer’s suggestion, we have evaluated the level of FM substitution based on PER, and relative items were added in the context.
Simply listing the Essential Amino Acids on a Table is not useful. Authors should estimate the respective Amino Acid indices and draw the relevant conclusions, i.e. Essential Amino Acid Index (Oser 1951); Chemical Score (Block-Mitchell 1946; Cowey & Sargent 1972). Which are the Essential Amino Acids requirements for this species at this stage, and how do they compare with the values listed in Table Three (3).
The Essential Amino Acid Index were calculated and no significant differences were observed. Additionally, we designed several feeding trials to evaluate the EAA requirements, in these studies we would evaluate the EAA indices with Essential Amino Acid Index.
How values of EAA in muscle protein listed in Table Five (5), do they compare with respective values of EAA for the same species in past research? Are they comparable and if not, why are they not comparable?
The values of EAA in muscle protein listed in this study could be comparable with those of EAA for the same species in past research, which were mainly due to the enormous difference of the experimental diets.
In the Discussion Section, the effect of use of PBPM inclusion IN OTHER species, is NOT of interest to us, for this manuscript and for this species. Although all species are fish, they are different taxonomic species and have different digestive activities and metabolism and are NOT comparable. You cannot compare the use and utilization of Protein and EAA between Onchorynchus spp. and Chana spp. and/OR Sparus aurata (mainly an omnivore) and/OR Epinephelus spp. (mainly a carnivore). The only thing you might be able to compare, could be the effect of PBPM inclusion in muscle composition and muscle organoleptic characteristics (taste, flesh texture or rigidity, colour). The serum blood and liver biochemical indices are useful in drawing conclusions and making comparisons with results of previous research, but for the SAME species and biological state (parr, smolt, etc).
According to reviewer’s suggestion, we revised it in the Discussion section.
Similarly, you cannot compare any results between Marine and Freshwater species! Due to differences in osmotic regulation mechanisms, their metabolism in NOT comparable!
According to reviewer’s suggestion, we rewrote the Discussion section.
3). Specific comments referring to line numbers, tables or figures that point out inaccuracies within the text or sentences that are unclear. [These comments should also focus on the scientific content and not on spelling, formatting or English language problems, as these will be addressed at a later stage by our internal staff.]
Following Reviewer’s suggestion, we have carefully corrected the errors and inadequacies in the text.
Once again, thank you very much for your comments and suggestions.

Round 2
Reviewer 2 Report
The revised version of “Partial replacement of fishmeal with poultry by-product meal in diets for coho salmon (Oncorhynchus kisutch) post-smolts”, by Yu et al.
I reviewed the revisions and the authors’ rebuttal statements. However, most of serious concerns I had suggested on the previous version have not been solved. So I bring the reviewing process to an end.
Broken-line analysis
The authors’ statements in the rebuttal are meaningless. Since the first line covering the lower levels of PBPM has a clear “negative” slope, it is not correct to regard the point of intersection with the other line covering the higher levels of PBPM as an optimum inclusion level of PBPM. Use one curve line by an appropriate regression analysis. Hassani et al. (2021) made the same mistake as the current authors, so that their study could not be nothing of justification. The value of square R = 1 may mean that the authors used average values of each treatment (n = 1) instead of individual values (n = 3), which is also not a correct manner.
Although the authors add a new graph (Figure 2) which presents the relationship between the level of PBPM and PER, the authors should bear in mind that PER decreases as fish grow. So, it is as a matter of course that the decreasing trend of PER between PBPM 0 and PBPM 20 is milder than that of SGR.
The conclusion of this work would be that PBPM is not a suitable ingredient as a partial replacement of fishmeal for coho salmon, which will be without doubt a useful finding. However, since the authors have refused to consider possible causative factors such as peroxidation of PBPM, this work is unfortunately not meaningful.
Feed intake, FCR and PER data
The authors’ rebuttal statements are non-responsive and incoherent. If the original feed intake data for calculation of FCR were on wet weight basis, then the dry feed intake data for calculation of FCR become different from the original feed intake data for calculation of PER. If the original feed intake data for calculation of PER are correct as the authors state, the revised FCR data then become incorrect. Such an uncertain rebuttal like the authors strongly reduces the reliability of all the data.
Quantification of the uneaten feed pellets
I am still unsure how the authors retrieved the pellets leftover on the bottom of “earthen pons” and how the authors identified the kind of pellets (treatment) leftover on the bottom of the pond?
Quantification of cystine
If cystine was destroyed during the acid digestion as the authors state, the data should be deleted from the results.
For example, L174, "antarctic" should be "Antarctic".